# Self-rated health, socio-economic status and post-traumatic stress disorder among couples from refugee backgrounds: Findings from the building a new life in Australia study

Thomas P. Nguyen[1,2] (iD), Pilar Rioseco[3], Sonia Terhaag[3] and Shameran Slewa-Younan[1,4,5]

[1]Mental Health, School of Medicine, Western Sydney University, Australia; [2]Victorian Transcultural Mental Health, St Vincent's Hospital Melbourne, Australia; [3]Australian Institute of Family Studies, Australia; [4]Translational Health Research Institute, School of Medicine, Western Sydney University, Australia and [5]Centre for Mental Health, The University of Melbourne School of Population and Global Health, Australia

**Keywords:**
couples; mental health; PTSD; refugee; self-rated health

**Corresponding author:**
Thomas P. Nguyen;
Email: thomas.nguyen@westernsydney.edu.au

## Abstract

Individuals from refugee backgrounds may experience higher rates of mental and physical health problems compared to the general population, yet the interdependence of these outcomes within couples remains poorly understood. This study aims to understand the relationship between post-traumatic stress disorder (PTSD), socio-economic status and self-rated general health (SRGH) among couples from refugee backgrounds living in Australia. Couples were nested within dyads using multi-level frameworks and mixed-effects logistic regression ($n = 436$ dyads). In respondents with likely PTSD, 61% of their partners were also likely to have PTSD compared to only 26% of partners in refugees with unlikely PTSD. After controlling for socio-economic factors, respondents with likely PTSD were significantly less likely to rate their health as 'excellent/very good' (OR = 0.20), compared to those with unlikely PTSD. Partners with likely PTSD were also less likely to rate their health as 'excellent/very good' (OR = 0.54). Individuals who were older, female, born in the Middle East, experienced less community support or more economic stressors were at greater risk of poorer SRGH. PTSD and SRGH had an interdependent effect within couples from refugee backgrounds. Familial and psychosocial contexts must be considered when developing health promotion and policies for refugee communities.

## Impact statement

Existing research has primarily examined the mental and physical health outcomes of refugees as individuals. This research demonstrates that post-traumatic stress disorder (PTSD) and self-rated general health not only affect refugees as individuals but have an interdependent effect within refugee couple dyads. In our study, refugees with PTSD were not only more likely to report poorer physical health, but their partners were also more likely to report greater PTSD symptoms and rate their own health as worse. These findings highlight the importance of considering familial and psychosocial contexts when shaping health promotion policies, clinical practice and support services for resettled refugees as well as incorporating a dyadic framework clinically in assessment and treatment. Approaches focus solely on individuals' risk overlooking the role of the family and couple systems which may influence physical health and well-being. Future research should build upon these findings and investigate the mechanisms and bidirectionality of this relationship as well as explore whether community-based social support and/or physical health programs may improve refugee health outcomes at both the individual and couple levels.

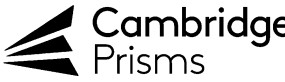



## Introduction

At the end of 2024, the number of forcibly displaced individuals worldwide reached a record high of 123.2 million people, of which 42.7 million are refugees (UNHCR, 2025). Refugees are individuals who have been forcibly displaced from their homelands due to conflict, persecution and/or human rights violations. Since the end of World War II, Australia has settled over 750,000 refugees, with the government granting 16,750 offshore resettlement visas in 2023–2024, which still represents a small proportion of the total number of people currently seeking asylum worldwide (Hugo, 2014; Affairs, 2024).

Individuals from refugee backgrounds are likely to have experienced one or more post-displacement stressors (e.g. unemployment and racial discrimination) which have been associated

with poorer mental health (e.g. PTSD, depression and anxiety) and self-rated physical health outcomes (Schweitzer et al., 2011; James et al., 2019). Alarmingly, studies set in Australia have also shown that the burden of poor mental and physical health often persists for several years following resettlement (Dowling et al., 2019; Nguyen et al., 2023).

Miller and Rasmussen's ecological model of refugee distress provides a framework for understanding why individuals from refugee backgrounds may be more susceptible to poorer mental and physical health outcomes (Miller and Rasmussen, 2017). The theory posits that distress experienced by refugees is shaped by not only pre-displacement traumas but also ongoing post-displacement stressors which occur across multiple levels of the social ecology including the family, community and macrosystem (Miller and Rasmussen, 2017). The continuous exposure to these various economic and social integration stressors has been shown to cause negative physical and psychological outcomes (Miller and Rasmussen, 2010). Moreover, these stressors are pervasive and thus likely to be experienced by families and specifically, couples of refugee backgrounds collectively. Given the majority of refugees resettled in Australia over the last decade come from collectivist cultural backgrounds, it is also important to recognise that their trauma appraisals are more likely to consider the effects of the trauma experienced on their role within their family or community over themselves (Engelbrecht and Jobson, 2016). In light of this, it is important that ongoing PTSD research in refugee populations acknowledges the role of the family system in the experience of individual trauma and post-displacement stressors.

Dyadic research looking at forcibly displaced couples has shown that PTSD symptoms in an individual may be associated with increased psychopathology reported by their partner (Mahmood et al., 2022; Khalil et al., 2023). One study sampled 101 Syrian and Iraqi couples from refugee backgrounds resettled in the US and found significant cross-associations between PTSD and depression/anxiety symptoms for both members of the couple dyad (Khalil et al., 2023). Another study sampled 687 Iraqi and Syrian refugee and internally displaced couples living in the Kurdistan Region of Iraq, and found traumatic experiences were positively associated with their own as well as their spouse's PTSD symptoms (Mahmood et al., 2022).

While prior research has shown the interdependence of PTSD with other psychopathology in forcibly displaced couples, no research to date has examined whether PTSD may be associated with poor physical health in forcibly displaced couples, despite strong evidence suggesting that PTSD is associated with dysregulated neurobiological stress pathways, chronic pain and poor cardiovascular health (Sareen et al., 2007; Pacella et al., 2013). For individuals from collectivist cultures, trauma is predominantly appraised as a physical stressor, with somatisation being a common feature of their post-traumatic stress (Engelbrecht and Jobson, 2016). When looking at research in individuals from refugee backgrounds specifically, one study showed better self-rated physical health was significantly associated with a lower number of PTEs while another study found PTSD symptoms moderate the inverse relationship between perceived adversity and perceived health (Nesterko et al., 2020; Javanbakht et al., 2022). However, it is important to note that these studies were set in Germany and the US, respectively, where physical and mental health supports for people of refugee backgrounds differ to Australia (Javanbakht et al., 2022).

Given these current gaps in the literature, this study uses the Actor–Partner Interdependence Model (APIM) to examine the associations between PTSD and self-rated general health in couples from refugee backgrounds who have resettled in Australia. Drawing on prior research, we hypothesised that partners of individuals from refugee backgrounds who reported PTSD symptoms would be more likely to report symptoms of PTSD themselves. Additionally, we hypothesised that partners of individuals from refugee backgrounds with PTSD symptoms would be more likely to report lower self-rated general health.

## Methods

### Participants and procedure

Data for this study were sourced from Wave 1 (October 2013 to March 2014) of the Building a New Life in Australia (BNLA) longitudinal study. The BNLA study aimed to understand the early resettlement outcomes of humanitarian migrants over the first five years of living in Australia and included those who arrived in Australia or were granted a permanent visa between May and December 2013. Extensive scoping work, including consultations with key stakeholders (e.g. settlement service providers and former humanitarian migrants), was undertaken to inform the content, design and methodology of the study (Studies, 2024). The study's surveys were also piloted with trained interviewers to assess for cultural sensitivity and understanding of the study materials (Studies, 2024). An overview of all outcomes assessed has been described in further detail previously (De Maio et al., 2014).

Participants who were potentially eligible for the study were randomly selected from one of 11 pre-identified study sites across Australia which included major cities and regional centres with high numbers of newly resettled humanitarian migrants. Principal applicants (PAs) were identified as either an individual or a representative of their family (87% male) and were sent a translated letter and brochure regarding the BNLA study. All PAs gave written informed consent to partake in the BNLA study before other family members, including spouses, were invited to participate. For the present study, heterosexual couples (married and de facto) from refugee backgrounds aged 18 years or older who completed Wave 1 of the BNLA study were included (434 couples, $n = 868$ individuals).

### Measures

Questionnaires and participant materials were translated into 14 languages by a professional translating company and were administered by trained bilingual interviewers. Professional interpreters were available for participants who spoke other languages.

PTSD was measured through the PTSD-8 which is an eight-item scale derived from the Harvard Trauma Questionnaire Part IV. The PTSD-8 assesses for the occurrence of PTSD symptoms across the three main symptom clusters (i.e. hypervigilance, intrusion and avoidance) over the past week using a 4-point Likert scale (not at all, rarely, sometimes and most of the time). It demonstrates strong psychometric properties and has also been validated in various populations, including humanitarian migrants (Hansen et al., 2010). For all Wave 1 participants, the PTSD-8 had a Cronbach's alpha of 0.92. Item 1 ('Overall, how would you rate your health during the past 4 weeks?') of the 8-Item Short Form Survey Instrument (SF-8) was used to measure self-rated general health. This item was scored on a 6-point Likert scale from Excellent to Very Poor (Ware et al., 1995). The SF-36 has been shown to demonstrate strong reliability and validity as a general measure of health status in various populations, including conflict-affected groups (Roberts et al., 2008; Lang et al., 2018).

Participants were asked about a range of different socio-demographic characteristics (e.g. age, sex, country of birth, level of education and current location), economic (English proficiency, economic stressors) and social integration factors (belonging, community support, loneliness). English proficiency was coded as 'no proficiency', 'low proficiency' and 'higher proficiency' based on the sum score of the participants' self-assessed proficiency in understanding, speaking, reading and writing (from 'not well' to 'very well' for each item). Participants were asked whether loneliness was a source of stress in the last 12 months and sense of belonging was assessed with the question 'Do you feel part of the Australian community' with response options ranging from 1 'always' to 5 'never'. Responses were reversed and used as a continuous variable in the analysis, with higher values indicating higher sense of belonging. Finally, financial hardship was assessed with the question 'In the last 12 months, has any of the following happened to you because you did not have enough money?' Six financial hardship items were collected and summed (e.g. not being able to pay bills on time, not being able to pay rent/mortgage on time and going without meals).

### Data analysis

To understand the role of own PTSD symptomatology and spouse's PTSD symptomatology on self-rated general health, we used the APIM (Kashy and Kenny, 2000). A key assumption of most analytical approaches is the independence of observations, which is violated in the study of couples. The APIM is a conceptual framework used to study relationships between dyads or groups, accounting for the lack of independence between group members (McCabe, 2020).

This analytical approach allows for dyadic relationships to be modelled. That is, the responses or outcomes for each individual within the dyad are modelled in the analysis accounting for the fact that they are not independent from each other (Cook and Kenny, 2005). We applied the APIM approach using a multi-level framework with two levels, where individuals (lower level) are nested within couples (upper level) and a random effect for each dyad is included in the model.

First, we performed a descriptive analysis of the distribution of PTSD and self-rated general health within couples, cross-tabulated by their spouse's PTSD and self-rated general health status. We then conducted multi-level random effects logistic regressions for binary outcomes to identify associations of own PTSD and spouse's PTSD with own self-rated general health (very good/excellent versus good/fair/poor/very poor self-rated general health). That is, own PTSD and spouse's PTSD are both independent variables in the model, with a random effect for each dyad accounting for the interdependence within couples. The models were constructed in steps in order to identify changes in regression coefficients as socio-demographic, economic and social integration factors were included in the model. First, we fitted an unadjusted model, including only own PTSD and spouse's PTSD. Then, socio-demographic characteristics were included in the model. As a third step, economic factors were added. Finally, social integration factors were included in the final model.

### Ethics approval

The BNLA study was approved by the Australian Institute of Family Studies Human Research Ethics Committee (protocol 13/03). Approval was gained from the Department of Social Services to access the BNLA study dataset.

### Results

The sample consisted of 436 couples from refugee backgrounds who had recently resettled in Australia. The main socio-demographic characteristics for the study sample are further described in Table 1. Only 25% of participants reported 'excellent' or 'very good' health (27% male, 23% female) with no differences by sex. A large proportion of participants reported likely PTSD (40% total, 39% male, 41% female) with no differences by sex either.

Among individuals from refugee backgrounds who reported having 'excellent' or 'very good' health, 36% had partners who also reported 'excellent' or 'very good' health. By contrast, only 13% of those reporting 'not very good' health had partners who reported having similarly high self-rated health. This association becomes even more pronounced when examining PTSD. Among individuals

**Table 1.** Socio-demographic characteristics

| Variable | Male %, mean (SD) | Female %, mean (SD) | Total %, mean (SD) |
|---|---|---|---|
| Age | 43 (11.9) | 38 (11.4) | 40 (12.0) |
| **Region of birth** | | | |
| Middle East | 62.4 | 63.3 | 62.8 |
| Afghanistan | 14.7 | 13.5 | 14.1 |
| South Asia | 15.4 | 15.6 | 15.5 |
| Other | 7.6 | 7.6 | 7.6 |
| **English proficiency** | | | |
| Not proficient | 33.5 | **42.4***  | 38.0 |
| Low proficiency | 48.6 | 41.3 | 45.0 |
| Higher proficiency | 17.9 | 16.3 | 17.1 |
| **Number of economic stressors (work, housing, finances)** | | | |
| 0 | 31.7 | 39 | 35.3 |
| 1 | 27.9 | 27.7 | 27.8 |
| 2 | 20.2 | 22.1 | 21.1 |
| 3 | 20.2 | 11.3 | 15.7 |
| **Education pre-displacement** | | | |
| No schooling | 12.7 | 16.4 | 14.5 |
| Up to 11 years school | 43.3 | 47.9 | 45.6 |
| 12 or more years education | 44.0 | **35.6***  | 39.8 |
| **Sense of belonging** | | | |
| Always | 48.3 | 44.5 | 46.4 |
| Most/some of time | 46.0 | 45.2 | 45.6 |
| Hardly ever/never | 5.7 | 10.3 | 8.0 |
| **Community support** | | | |
| High support | 24.1 | 28.8 | 26.4 |
| Some support | 35.4 | 35.8 | 35.6 |
| No support | 40.5 | 35.4 | 37.9 |
| **Loneliness** | | | |
| No | 92.3 | 85.4 | 88.8 |
| Yes | 7.7 | **14.6***  | 11.2 |

Note:
*indicates a statistically significant difference by sex $p < 0.05$.

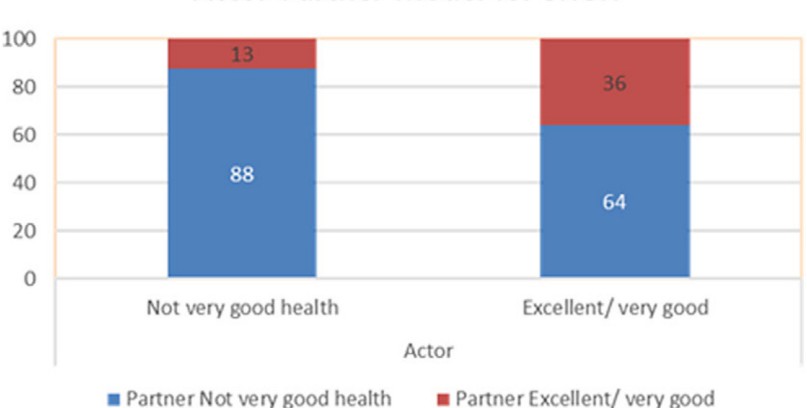

**Figure 1.** Actor–partner interdependence model for self-rated general health.

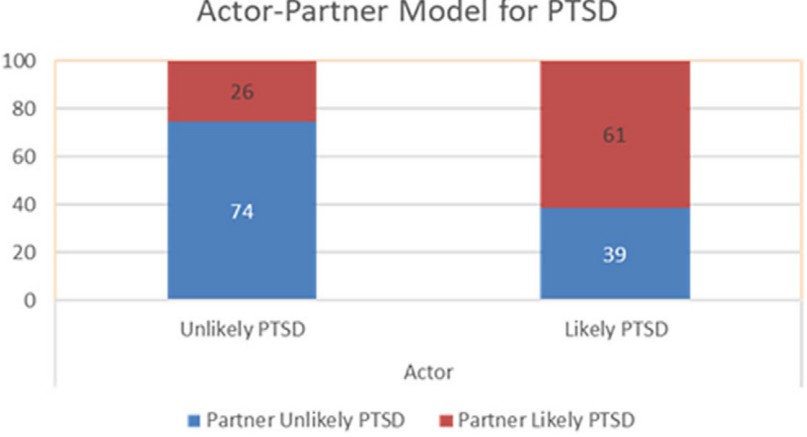

**Figure 2.** Actor–partner interdependence model for PTSD.

classified as having likely PTSD, 61% had partners who were also classified as likely PTSD, whereas only 26% of partners of individual classified as unlikely to have PTSD reported likely PTSD. These results are depicted in Figures 1 and 2.

Table 2 shows the mixed-effects logistic regression models for those reporting 'excellent' or 'very good' health unadjusted and adjusted for socio-demographic, economic and social integration factors. Respondents' own PTSD and spouses' PTSD were significantly associated with respondents' own self-rated general health in the unadjusted model, as well as in the models including socio-demographic, economic and social integration factors. In the full model, the effect size for the association of own PTSD with self-rated general health (OR = 0.2) was equivalent to a fivefold increase in the odds of having poorer self-rated general health, compared with respondents without PTSD. The association for spouse's PTSD (OR = 0.54) in the full model was equivalent to a 1.8-fold increase in the odds of the individual having poorer self-rated general health, compared with respondents whose spouses did not meet criteria for PTSD.

In terms of socio-demographic characteristics of the respondent, age (older respondents were less likely to report 'excellent' or 'very good' health compared with younger respondents), sex (females were less likely to report 'excellent' or 'very good' health compared with males) and region of birth (respondents born in Afghanistan and 'other' countries were more likely to report good/excellent health

compared with those born in the Middle East) were significantly associated with own self-rated general health, with little variation when additional factors were included in the model. In terms of economic factors, respondents with a larger number of economic stressors were less likely to report good/excellent self-rated general health (OR = 0.78). Those who received some community support compared to high community support were also less likely to report 'excellent' or 'very good' health (OR = 0.55).

The intraclass correlation in the multi-level models decreased from 0.51 in the unadjusted model to 0.30 in the fully adjusted model. That is, the proportion of the variance in self-rated general health accounted for by the couple decreased as additional factors were included in the model, showing that the shared variance within couples was partially explained by socio-demographic, economic and social integration factors controlled for in the models. An interaction effect was included in the final model between spouse's PTSD and respondent's gender. The interaction term was not significant indicating that gender does not moderate the effect of spouse's PTSD on own self-rated general health (*p*-value = 0.660).

### Discussion

This paper sought to understand the relationship between the PTSD symptomatology and self-rated general health among couples from refugee backgrounds who were recently resettled in

**Table 2.** Mixed-effects logistic regression models

| Independent variables | Unadjusted (n = 824) OR | Socio-Demographic (n = 824) OR | Economic (n = 814) OR | Social Integration (n = 761) OR |
|---|---|---|---|---|
| Own PTSD | 0.13*** | 0.18*** | 0.20*** | 0.20*** |
| Spouse PTSD | 0.35*** | 0.46** | 0.49** | 0.54** |
| Age | | 0.94*** | 0.95*** | 0.95*** |
| Female | | 0.52** | 0.51** | 0.61* |
| Region of birth | | | | |
| Middle East (reference category) | | 1.00 | 1.00 | 1.00 |
| Afghanistan | | 3.43** | 3.70*** | 4.38*** |
| South Asia | | 0.74 | 0.56 | 0.48 |
| Other | | 8.11*** | 7.55*** | 5.84*** |
| Language proficiency | | | | |
| Not proficient (reference category) | | | 1.00 | 1.00 |
| Low proficiency | | | 1.64 | 1.66 |
| Higher proficiency | | | 2.01 | 1.91 |
| Number of economic stressors | | | 0.74* | 0.78* |
| Sense of belonging | | | | |
| Always (reference category) | | | | 1.00 |
| Most/ some of the time | | | | 0.90 |
| Hardly ever/ never | | | | 0.66 |
| Received community support | | | | |
| High support (reference category) | | | | 1.00 |
| Some support | | | | 0.55* |
| No support | | | | 0.66 |
| Loneliness (ref. no loneliness) | | | | 0.45 |
| Random effects couple variance | 3.42 | 2.23 | 2.08 | 1.41 |
| Couple ICC | 0.51 | 0.40 | 0.39 | 0.30 |

*Note*: ICC, intraclass correlation; OR, odds ratio. *p < 0.05, **p < 0.01, ***p < 0.001.

Australia. We found a high level of interdependence of PTSD symptoms within couples from refugee backgrounds. Moreover, to our knowledge, this is the first study to demonstrate an interdependent relationship between PTSD and self-rated general health within couple dyads from refugee backgrounds.

Our findings showed a high predominance of PTSD symptoms in individuals from refugee backgrounds whose partners also displayed symptoms of PTSD, which aligns with findings from previous research conducted in refugee populations (Mahmood et al., 2022; Khalil et al., 2023). For individuals from refugee backgrounds, these findings suggest that the presence of PTSD symptoms in one partner may increase the vulnerability of PTSD symptoms in the other; however, the directionality of this relationship remains unclear. The ecological model of refugee distress highlights several post-displacement stressors at different levels of the social ecology, some of which may affect couples more saliently such as family separation, housing instability and financial insecurity (Miller and Rasmussen, 2017). These may be exacerbated by trauma appraisals in which individuals may consider the impact of trauma on their partner or their perceived role within the relationship, rather than on themselves. Further research exploring the mechanisms underlying the intra-couple relationship of PTSD symptoms may wish to

specifically examine the role of post-displacement stressors and differing trauma appraisals.

We also found that individuals from refugee backgrounds with greater PTSD symptoms reported poorer self-rated general health, extending upon previous research showing that self-rated physical health declines as the number of PTEs experienced increases. Moreover, our study is the first to show that PTSD and self-rated physical health may be interdependently related within the couple dyad for individuals from refugee backgrounds. These findings may reflect trauma appraisals that are more common among refugee groups from collectivistic cultures whereby a partner's PTSD symptoms may be perceived as disruptive to the family unit and exacerbate one's own physical health complaints. Future research should examine the directionality of this association and whether poorer self-rated physical health may be secondary to the somatisation of trauma-related symptoms.

With respect to specific demographic factors, our findings that older age, being female, those born in the Middle East, those reporting more economic stressors and those reporting less community support were groups that were more likely to report poor self-rated general health align with previously published cross-sectional and longitudinal research (Dowling et al., 2019;

Haj-Younes et al., 2020; Ambrosetti et al., 2021). Longitudinal studies have shown that while there are static risk factors that may predict poor self-rated general health long-term (e.g. female, older age), there are more dynamic predictors which may be modifiable such as economic stressors and social support (Dowling et al., 2019; Haj-Younes et al., 2020). Future research should thus endeavour to evaluate the effect of community-based social support and/or physical health programs that target these modifiable risk factors and understand whether they may help improve self-rated general health in forcibly displaced populations. Designing community-based, scalable interventions that are culturally safe are especially important given the paucity of preventative public health care services in many low- and middle-income countries that have been decimated by years of conflict.

This study has a few limitations that should be noted. First, the cross-sectional design of this study precludes any assertions of causal associations between PTSD and self-rated general health. While most primary respondents in our study were male, gender imbalances are unfortunately a common issue in studies with forcibly displaced populations and thus, future studies of this scale should attempt to address these sampling biases. Second, the use of a self-report, non-diagnostic tool to measure PTSD may have resulted in an overestimation of the prevalence of PTSD within our sample. However, it is important to note that the PTSD-8 scale was chosen in consideration of reducing the significant burden of other questionnaires used in the study. Moreover, no information on pre-displacement physical health conditions or trajectories in health status pre- and peri-displacement were collected which limits our ability to control for the impact of pre-displacement experiences on the association between general health and PTSD. Lastly, the use of a single-item measure for self-rated general health limits a broader and more holistic understanding of the overall physical health of our sample but was partly chosen due to its strong reliability and validity especially when considering its brevity. Notwithstanding these limitations, our study has several strengths. These include the use of a nationally representative and large sample size of humanitarian migrants from diverse backgrounds as well as the use of regression modelling to control for multiple variables along with the use of APIM to perform couple-level analyses. Our study is also the first of its kind to explore the relationship between PTSD, socio-economic status and self-rated general health within couples from refugee backgrounds resettled in a high-income country.

## Conclusions

Our study highlights the interdependent effect of PTSD and self-rated general health within couples from refugee backgrounds who have newly resettled in Australia. Economic and social integration stressors were also found to be important determinants of self-rated general health. Our findings further reinforce the growing recognition that supporting these communities requires a collectivist approach, with health promotion and psychological treatment strategies that consider couples, families, the wider community and their broader psychosocial contexts. Clinically, our findings underscore the importance of assessing PTSD and physical health within a dyadic framework, given a partner's symptoms may influence other's symptomology. Our findings further suggest the potential values of couples-based interventions for improving physical and mental health outcomes in individuals from refugee backgrounds. Future research should examine the underlying mechanisms and directional pathways underlying these associations, with particular attention to collectivistic trauma appraisals and post-displacement stressors.

**Open peer review.** To view the open peer review materials for this article, please visit http://doi.org/10.1017/gmh.2026.10187.

**Data availability statement.** The BNLA dataset used in the present study is publicly available to researchers who have obtained permission from the Australian Government Department of Social Services.

**Acknowledgements.** This paper uses unit record data from the Building a New Life in Australia study conducted by the Australian Institute of Family Studies for the Australian Government Department of Social Services (DSS) (Department of Social & Australian Institute of Family, 2019). The findings and views reported in this paper are those of the authors and should not be attributed to the Australian Government, DSS or any DSS contractors or partners.

**Author contribution.** PR and ST conceived the study and conducted the formal analysis. TPN and PR wrote the original draft. All authors reviewed and edited the manuscript prior to submission.

**Financial support.** No funding was obtained for this study.

**Competing interests.** The authors declare none.

**Ethics statement.** The BNLA study was approved by the Australian Institute of Family Studies Human Research Ethics Committee (protocol 13/03). Approval was gained from the Department of Social Services to access the BNLA study dataset.

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
