## [Reviewer Report]

Thank you to the authors for their effort and for addressing such an important and significant topic. However, given the methodological limitations and lack of contextual consideration I believe the paper does not meet the standards required for publication at this stage and require major revisions. Please see below some general comments as well as some specific ones.

General Comments

• The term “refugees” is not recommended in contexts where individuals may no longer legally hold refugee status, even though it is part of their lived experience. More appropriate alternatives include “people from refugee backgrounds” or “an individual from a refugee background.”

• The introduction lacks coherence and does not clearly establish the background or significance of the study based on existing literature. It moves between PTSD, general health, and partner experiences without a structured argument explaining why this topic matters.

o What is the relevance of this research to the day-to-day experiences of people from refugee backgrounds in Australia?

o What challenges do couples face regarding PTSD and health in this context that make this research significant?

o How are these concepts potentially related? For example:

• If PTSD symptoms decrease, does health improve?

• If one partner’s symptoms reduce, does the other partner’s health benefit?

At present, the connection between PTSD and health outcomes is unclear.

• There are numerous references to veteran experiences, which are contextually irrelevant to PTSD among people from refugee backgrounds. PTSD cannot be understood without considering the context of trauma. There is extensive research on PTSD in refugee populations that should be prioritized.

Abstract

The abstract should begin with a clear, concise opening sentence. For example, the current sentence:

“Refugees are more likely to report mental and physical health problems, yet how these factors may be interdependently related to each other remains poorly understood”

is vague. Compared to whom? In what context?

Introduction

• Australia’s intake of refugees is not modest compared to global numbers—it is a very small proportion of the 35.3 million displaced people worldwide. Updated 2024 figures are available on the UNHCR website and should be used.

• This sentence does not flow logically:

“Concerningly, refugees report poorer self-rated general health compared to immigrant populations and this trend appears to persist over the first few years following resettlement (10).”

Consider clarifying its relevance to the argument.

• Page 5, first paragraph: Discussing social determinants of health without context is problematic. For example, the U.S. health system differs significantly from Australia’s, and these differences influence outcomes. If citing U.S.-based research, explain its relevance despite these contextual differences.

• “However, a growing body of research has begun to investigate the interdependent effects of PTSD in couple dyads, particularly in veteran populations (12).”

This is misleading. The cited studies focus on U.S. military veterans—individuals who engaged in armed conflict, sometimes involving aggression and violence. Their PTSD experiences differ fundamentally from those of survivors of violence. Using this comparison risks misrepresenting refugee experiences.

Analysis and Discussion

• The statement “We not only found that PTSD symptoms were related to poor self-rated general health within refugee individuals themselves but there was an interdependent relationship within the couple dyad” and some other similar statements raises questions:

o The analysis claims that PTSD symptoms are related to poor self-rated general health and that there is an interdependent relationship within couple dyads. However, this interpretation overlooks critical contextual factors. For example, individuals from refugee backgrounds often spend years in war or refugee camps with limited access to food, healthcare, and basic necessities, which significantly impacts their health independent of PTSD. Ignoring these historical and structural determinants oversimplifies the relationship and risks drawing misleading conclusions. Additionally, if participants were newly resettled, how did the authors ensure they understood the questionnaires and concepts like PTSD in a culturally informed way? Simply using interpreters does not guarantee comprehension or informed responses. Without clear evidence of culturally appropriate psychoeducation and rigorous procedures to ensure validity and reliability, the data and subsequent conclusions cannot be considered robust. These methodological gaps undermine the credibility of the study and its contribution to the literature.

---

## [Reviewer Report]

Thank you for the opportunity to review this manuscript. This study investigates the probable PTSD and self-rated poor health among refugee couples, considering the interdependent nature of their mental and physical health. It is commendable that the research team focused on dyads, given the predominance of studies taking an individualistic approach. I believe this paper has important implications for research and practice and will be a valuable addition to the collection of Cambridge Prism Global Mental Health. Before suggesting its publications, I have major revision requests, focusing on the following points.

Abstract:

1. The first sentence of the Abstract sounds like the aim of the paper would be to investigate the bidirectional relationship between PTSD and self-rated health. I suggest revising this sentence to explicitly indicate that the authors are investigating the interdependency within couples, not between mental and physical health.

Introduction:

1. Although the title and abstract indicate that socio-economic stressors are one of the main focuses of the paper, there is no mention of these stressors and their impact on mental and physical health in the Introduction. The authors only very briefly say that post-migratory stressors may be a stronger predictor of mental health on Page 4, Line 35-40.

2. The authors start their Introduction with the impact of traumatic experiences; however, this is not one of the focuses of the paper, and as the narrative is built on traumatic experiences- negative health outcomes- postmigration stressors- negative health outcomes, I would expect to see that this important variable would have been investigated as a risk factor in this study.

3. I find this sentence a bit confusing as it implies that the authors examined the interdependent relationship between PTSD and health, Page 5, Line 10-15. Thus, further research needs to elucidate whether PTSD and self-rated general health interdependently affect refugees and more importantly, whether there are other social determinants that may explain these effects.

4. Overall, the Introduction is descriptive, attempting to provide the link between the key study variables. I think it can benefit greatly from a theoretical framework explaining why the key stressors (traumatic experiences and post-migration stressors) impact overall health within couples. I suggest that the authors to either integrate an ecological model of refugee distress or a theory/conceptual framework suggesting the interdependency within couples.

Method:

1. More information on the analytical approach (Actor Partner Interdependence Model) is needed. The authors provided only a brief description of the overall aim of this approach.

2. Page 9, Line 32-33: That is, own PTSD and spouse’s PTSD are both independent variables in the model. Does “independent” need to be replaced with “interdependent”?

3. The authors included both socio-economic and integration stressors in their analyses; however, there is no mention of these variables in the Introduction.

4. Also, how do the authors define a couple? Is it married couples?

Results:

1. Page 13, Line 32-35: Those who received some social support compared to high support were also less likely to report ‘excellent’ or ‘very good’ health (OR = 0.55). I’m not sure what the authors refer to as social support here. Is it loneliness or belonging item that they used in their analyses?

Discussion:

1. The Discussion could significantly benefit from a deeper discussion on the key findings. For instance, the finding that the spouse’s PTSD is associated with a higher risk for poor health than the participant’s own PTSD is of great importance to discuss. What does this result mean?

2. What does it mean that those whose partners with probable PTSD were more likely to have PTSD? There is no discussion on this finding.

3. Given the importance of the current findings, I found that the study’s clinical and research implications are not well-articulated. The authors suggested some general suggestions on how to address modifiable risk factors based on their findings related to self-rated general health. The main findings’ implications were only mentioned in one sentence in the Conclusion Section.

Overall, I believe this is an important paper, shedding light on a relatively underexplored aspect of refugee mental health.

---

## [Editor Report]

Thank you for submitting this manuscript on an important topic. Both reviewers recognize the potential contribution of examining the mental and physical health interdependence within couples from refugee backgrounds. However, they identified conceptual, contextual, and methodological limitations that require further consideration and revision. In particular, the introduction introduces several concepts that are related, but not the focus of the paper. As noted by one of the reviewers, improving the coherence and focus of the introduction and its connections to existing theoretical frameworks would improve the manuscript. Additionally, I encourage the authors to consider the recommendation about terminology. Additionally, the reviewers requested some clarifications about the methodology. I agree with the reviewers that this study makes an important contribution and encourage the authors to consider submitting a revised version of the manuscript that takes into account some of these recommendations.

---

## [Reviewer Report]

Authors sufficiently addressed most of my points. I would like to aske for further clarification on the below points.

• We have now elaborated on the impact of these stressors on mental and physical health, “Individuals from refugee backgrounds are likely to have experienced one or more potentially traumatic events (PTEs) either pre-migration (e.g., exposure to violence or torture), peri-migration (e.g., family separation, human trafficking) or post-migration (e.g., financial stress, social integration stressors such as discrimination) (4). Exposure to these PTEs has been associated with poorer mental health (e.g., post-traumatic stress disorder, depression) and self-rated physical health in individuals from refugee backgrounds (5, 6). Studies set in Australia have also shown that the burden of poor mental and physical health often persists for several years following resettlement (7, 8).”

Reply: I’m not sure if I followed this correctly. Do the authors include postmigration stressors as potentially traumatic events? It is well-established in the literature that post-displacement stressors have their own category despite the possibility that they can be traumatic in nature. Also, I suggest the authors to revise the use of post-migration stressors as post-displacement stressors.

• We have now more clearly linked traumatic experiences to post-traumatic stress and expanded more on post-migration stress as well as its links to negative health outcomes. “Exposure to these PTEs has been associated with poorer mental health (e.g., post-traumatic stress disorder, depression) and self-rated physical health in individuals from refugee backgrounds (5, 6)…distress experienced by refugees is not only shaped by pre-migration traumas, but also by ongoing post-displacement stressors which occur across multiple levels of the social ecology including the family, community and macrosystem (9). The continuous exposure to these various socio-economic and social integration stressors has been shown to cause negative physical and psychological outcomes (11).

Reply: I’m not sure if my point has been addressed fully here. I’m wondering if the traumatic experiences are included in the analysis given the emphasis in the Introduction.

I also suggest that the authors use either “economic stressors” or “socio-economic stressors”. The “socio” component of socio-economic already encompasses social dimensions, so the current phrasing may be conceptually redundant.

• We have incorporated the ecological model of refugee distress as a framework to explain why traumatic experiences may impact overall health, “Miller and Rasmussen’s ecological model of refugee distress provides a framework for understanding why individuals from refugee backgrounds may be more susceptible to poorer mental and physical health outcomes (9, 10). The theory posits that distress experienced by refugees is not only shaped by pre-migration traumas, but also by ongoing post-displacement stressors which occur across multiple levels of the social ecology including the family, community and macrosystem (9). The continuous exposure to these various socio-economic and social integration stressors has been shown to cause negative physical and psychological outcomes (11).”

Reply: There is a reference to Hobfoll in Miller and Rasmussen’s model- please make sure that the references are updated and relevant.

---

## [Editor Report]

Thank you for resubmitting your manuscript to Global Mental Health. One reviewer has requested additional clarification and possible revision on two of their original suggestions as well as a correction to one of the citations. We agree with these points and hope the authors will consider revising the manuscript accordingly.